# Time-restricted feeding reduced blood pressure and improved cardiac structure and function by regulating both circulating and local renin-angiotensin systems in spontaneously hypertensive rat model

Xin Yi[1,2], Razif Abas[3], Raja Abdul Wafy Raja Muhammad Rooshdi[1], Jie Yan[2], Canzhang Liu[2], Chongshuang Yang[1], Teng Gao[2], Weijing Sun[1], Ummi Nadira Daut[1]*

1 Department of Internal Medicine, Faculty of Medicine and Health Sciences, Universiti Putra Malaysia, Serdang, Selangor, Malaysia, 2 Department 1 of Cardiovasology, North China University of Science and Technology Affiliated Hospital, Tangshan City, Hebei Province, China, 3 Department of Human Anatomy, Faculty of Medicine and Health Sciences, Universiti Putra Malaysia, Serdang, Selangor, Malaysia

* umminadira@upm.edu.my

**Data availability statement:** All relevant data are within the manuscript and its Supporting Information files.

**Funding:** The author(s) received no specific funding for this work.

## Abstract

### Objective

To investigate whether time-restricted feeding (TRF) can reduce blood pressure (BP) and improve cardiac structure and function in spontaneously hypertensive rats (SHRs) by regulating the renin-angiotensin system (RAS).

### Methods

Wistar Kyoto rats and SHR underwent 16 weeks of TRF intervention, with daily feeding restricted to 9 am–5 pm. The effects of TRF on systolic BP, diastolic BP, mean BP, body weight (BW), heart weight (HW), HW/BW ratio, cardiac structure and function, and RAS activity in the circulating and left ventricular (LV) tissues were investigated.

### Results

TRF effectively reduced systolic BP, mean BP, diastolic BP, and BW; improved hypertension-induced cardiac structural and functional damage; and inhibited the ACE-Ang-II-AT1 axis in circulating and LV tissues.

### Conclusion

TRF effectively inhibits RAS activity in both circulating and LV tissues, thereby lowering BP and mitigating structural and functional cardiac damage associated with hypertension.

**Competing interests:** The authors have declared that no competing interests exist.

**Abbreviations:** TRF, Time-Restricted Feeding; BP, Blood Pressure; SBP, Systolic Blood Pressure; DBP, Diastolic Blood Pressure; MBP, Mean Blood Pressure; BW, Body Weight; HW, Heart Weight; LV, Left Ventricle; RAS, Renin-Angiotensin System; ACE, Angiotensin-Converting Enzyme; Ang-II, Angiotensin II; AT1, Angiotensin II Type 1 Receptor; ANP, Atrial Natriuretic Peptide; BNP, B-type Natriuretic Peptide

# 1 Introduction

Hypertension remains a significant global public health issue associated with substantial morbidity and mortality. The current prevalence of hypertension is approximately 33% among individuals aged 30 to 79 years worldwide [1]. Hypertension is a leading risk factor for global mortality, accounting for 19% of all deaths. Additionally, more than 50% of cardiovascular disease-related deaths can be attributed to hypertension, particularly among patients with uncontrolled high blood pressure (BP) [2,3]. Persistently BP can lead to structural and functional damage to the heart, resulting in serious complications [4,5]. There is a lack of a definitive cure for hypertension, and disease progression can only be mitigated through medication and other approaches. Emerging evidence suggests that a healthy diet plays a pivotal role in the treatment and prevention of hypertension [6]. Consequently, hypertension and its induced cardiac remodeling are clinically challenging to manage.

Intermittent fasting(IF) is a novel dietary intervention that includes alternate-day fasting (ADF), time-restricted feeding (TRF), and other variations alternating between regular fasting and free feeding periods [7]. It is effective in delaying the onset and progression of various chronic diseases, including cardiovascular disease [8]. A recent study [9] demonstrated that 4 weeks of ADF reduced BP and ameliorated cardiac hypertrophy in mice with metabolic heart disease induced by high-fat or high-sugar diets. These beneficial effects of ADF may be related to its local inhibition of the renin-angiotensin system (RAS) in the cardiac tissues. However, hypertension-induced cardiac remodeling is not consistent with the pathological process of metabolic heart disease [10,11]. Furthermore, the study primarily assessed cardiac morphology, neglecting the functional aspects. Both TRF and ADF are commonly used IF methods, although TRF has a shorter fasting period and is more similar to the normal human diet, potentially resulting in better patient compliance. However, it remains unclear whether there is a difference in outcomes between these two IF methods.

Recent experimental studies on TRF have mainly focused on obesity and metabolism [12–15]. Although some studies have confirmed) that TRF can lower (BP) and improve its circadian rhythm in different animal models [16,17], the underlying mechanism remains unclear. RAS is a central component of the pathological process of hypertension and hypertension-induced cardiac remodeling and is a key therapeutic target [18]. Angiotensin II (Ang-II), the primary effector molecule of the RAS, exerts its effects by binding to the angiotensin II Type 1 (AT1) receptor, leading to vascular smooth muscle contraction and elevated BP [19]. The RAS plays a pivotal role in cardiovascular remodeling. Ang-II and aldosterone induce myocardial hypertrophy, fibroblast proliferation, and activation of inflammatory immune cells through hemodynamic alterations, as well as direct growth and proliferative effects. These mechanisms contribute to left ventricular (LV) hypertrophy and cardiac remodeling. Prolonged hypertension further exacerbates these pathological processes, ultimately progressing to heart failure [20,21]. There are two distinct RAS in the circulation and cardiac tissues. The circulating RAS primarily regulates short-term blood flow, while the tissue RAS is mainly related to organ remodeling. These two systems operate independently of each other but work together to regulate cardiovascular system homeostasis [22,23].

Currently, there is a gap in the research concerning the association between TRF and hypertension-induced cardiac remodeling. We hypothesized that TRF could reduce BP and ameliorate hypertension-induced cardiac remodeling by inhibiting both the local and circulating RAS. Therefore, our study aimed to evaluate the effects of TRF on BP levels in hypertension-induced cardiac remodeling and the RAS in spontaneously hypertensive rats (SHR). The study underscores non-pharmacological interventions in hypertension treatment, offering potential implications for clinical practice and novel therapeutic strategies.

Our research describes the mechanisms of TRF's beneficial effects, providing a foundation for future studies on dietary interventions as adjunctive therapies for hypertension.

## 2  Materials and methods

### 2.1  Laboratory animals

Twenty-four 7-week-old male rats, including SHR ($n = 12$) and Wistar-Kyoto (WKY) rats ($n = 12$), each weighing 200–220 g, were selected. All rats were of specific pathogen-free grade and acquired from Beijing Huafukang Technology Co. Ltd. (Beijing, China Animal License No. SCXK(Beijing) 2019-0008). The rats were housed in a quiet feeding room with a controlled room temperature ($22 \pm 2$) °C and humidity maintained at ($60 \pm 5$)% under a 12-hour light-dark cycle. The experiment commenced after 1 week of adaptive feeding. After the adaptive feeding period, BP parameters, including systolic BP (SBP), diastolic BP (DBP), and mean BP (MBP), were measured using non-invasive monitoring in all rats to evaluate the establishment of the SHR hypertension model. SHR rats who had a significant elevation in SBP, with a statistically significant difference compared to WKY rats ($P < 0.05$) were included. Animals with other illnesses or abnormalities (e.g., infections or abnormal behavior) and animals that died during the adaptive feeding period or failed to meet the BP criteria were excluded. The protocol was approved by the Institutional Animal Care and Use Committee (IACUC) of Kangtai Medical Laboratory Service Hebei Co., Ltd. (Hebei province, China), with approval number MDL2023-08-30-01. All experiments were conducted in compliance with the National Institutes of Health guidelines for the use of experimental animals. After the adaptive feeding, SBP levels were higher in all SHR rats compared to all WKY rats (148.349 ± 8.176 mmHg vs. 178.604 ± 10.937 mmHg, $P < 0.001$), suggesting that hypertension had already developed in the SHR rats. All rats were included in the study. All WKY rats were randomly assigned to the WKY-NON-FASTING group ($n = 6$) and WKY-TRF group ($n = 6$), while all SHR were randomly divided into the SHR-NON-FASTING group ($n = 6$) and SHR-TRF group ($n = 6$).

### 2.2  Time-restricted feeding intervention

TRF intervention was applied to the WKY-TRF and SHR-TRF groups. The rats in these groups were strictly allowed to feed between 9 am and 5 pm each day, with no access to food outside these hours.Because the method is more in line with normal human eating habits. All the rats had unrestricted access to water. The rats in the WKY-NON-FASTING and SHR-NON-FASTING groups were permitted to feed and drink water freely at all times. There were no restrictions on the total amount of food consumed by the rats, and all were fed a standard diet. The intervention was concluded at 24 weeks of age.

### 2.3  Continuous blood pressure monitoring

Continuous BP monitoring, including SBP, DBP, and MBP, was performed weekly during the TRF intervention period. The thermostatic non-invasive BP measurement device (XH200, Beijing Zhongshi Dichuang Technology Development Co., Ltd., Beijing, China) and Biosignal Acquisition and Processing System (MADLAB-4C/501H, Beijing Zhongshi Dichuang Technology Development Co., Ltd., Beijing, China) were used for the measurements. BP measurements were taken when the rats were calm. An assistant secured the rats on the measuring device and recorded the measurement sequence. The operator conducted three consecutive measurements, calculated the average, and documented the results.

## 2.4 Euthanasia, tissue collection, and hematoxylin and eosin staining

On euthanasia day, all rats were fasted for 6 h and then anesthetized with 1.25% tribromoethanol (150 mg/kg). After anesthesia, surgical scissors were used to open the abdominal cavity and cut the ribs in a straight line to extract blood through the aorta. The blood was then centrifuged and stored in a refrigerator at -80°C for subsequent use. Simultaneously, the chest was quickly opened, and the heart was removed. Cold saline was used to flush blood from the heart, excess vascular tissue was removed, and the excess water was blotted out with filter paper. The heart was photographed, and heart weight (HW) was determined. LV tissues were then isolated. Some LV tissues were fixed with 4% formaldehyde, paraffin-embedded, and cut into 4-μm-thick cross-sections by microtome (RM2235, Leica, Germany). Hematoxylin and eosin (HE) staining was performed to observe the micro-structure of cardiomyocytes under a light microscope (DM3000, Leica, Germany) at 200X magnification. The remaining LV tissues were stored in a refrigerator at -80°C for subsequent use.

## 2.5 Real-time quantitative reverse transcription polymerase chain reaction (RT-qPCR)

The mRNA expression levels of atrial natriuretic peptide (ANP) and B-type natriuretic peptide (BNP) in the LV tissues were detected using RT-qPCR to evaluate cardiac function. First, mRNA was extracted from LV tissues using TRIzol (RN0102, Aidlab Biotechnologies, Ltd., Beijing, China). mRNA was synthesized using a SuperScript III Reverse Transcriptase Kit (A502, Exongen, Chengdu, China). RT-qPCR experiments were then conducted using SYBR qPCR mix (4472920, ABI-Invitrogen,California, USA) and RT-qPCR System (Q2000B, Longgene, Hangzhou, China). Relative mRNA expression was quantitatively analyzed by 2-ΔΔCT calculated as follows: ΔCT (sample to be tested) = CT value of the target gene of the sample to be tested - CT value of the loading control gene of the sample to be tested; ΔCT (control sample) = CT value of the target gene of the control sample - CT value of the loading control gene of the control sample; ΔΔCT = ΔCT (sample to be tested) - ΔCT (control sample). All primers used were synthesized by Beijing Bomaide Gene Technology Co., Ltd. (Beijing, China). The primer sequences are listed in Table 1.

## 2.6 Enzyme-linked Immunosorbent Assay (ELISA)

Angiotensin-converting enzyme (ACE), Ang-II, and AT1 levels in the serum and LV tissues were measured using an ACE ELISA kit (MD13382, MDL, Beijing, China), Ang-II ELISA kit (MD11247, MDL, Beijing, China), and AT1 ELISA kit (FY-A014619, FUYUBIO, China), respectively, to assess the activity of RAS. The procedure was carried out according to the manufacturer's instructions. Optical density was measured using a microplate reader (Model

**Table 1. Primers sequences of RT-qPCR.**

| Primers | Direction(5'- 3') | Sequences |
|---|---|---|
| ANP | Forward | TGGACCCCTCCGATAGATCTGC |
| | Reverse | CGCTCTGGGCTCCAATCCTG |
| BNP | Forward | CTCTCAAAGACCAAGGCCCTA |
| | Reverse | GCAGCTTGAACTATGTGCCATC |
| Actin | Forward | CTGAACGTGAAATTGTCCGAGA |
| | Reverse | TTGCCAATGTGATGACCTG |

ANP: Atrial natriuretic peptide; BNP: B-type natriuretic peptide.

680, BIO-RAD, California, USA). A standard curve was plotted using Word Processing System-Excel (version 12.1.0 16929, Kingsoft, Beijing, China), and sample concentrations were calculated for quantitative analysis.

## 2.7 Statistical analysis

Statistical data analysis was performed using SPSS 28.0 software (IBM, USA). All data were expressed as mean ± standard deviation of the mean. One-way ANOVA was used for between-group comparisons, and the least significant difference test was used for within-group comparisons. Statistical differences were defined when the P value was less than 0.05 ($P < 0.05$).

# 3 Results

## 3.1 Continuous BP levels monitoring

At 8 weeks of age, SBP, DBP, and MBP levels were higher in the two SHR groups compared to the two WKY groups, suggesting that hypertension had already developed in the SHR before the TRF intervention. From age 8 to 10 weeks, the SBP, DBP, and MBP levels in the SHR-NON-FASTING and SHR-TRF groups showed an upward trend. However, at 11 weeks, SBP, DBP, and MBP levels in the SHR-TRF group significantly decreased yet continued to increase in the SHR-NON-FASTING group. At this time point, SBP, DBP, and MBP levels were significantly lower in the SHR-TRF group compared to the SHR-NON-FASTING group but still higher than those in both the WKY-NON-FASTING and WKY-TRF groups ($P < 0.05$)This difference persisted from week 2 to week 16, continuously. No significant differences in SBP, DBP, and MBP were observed between the WKY-NON-FASTING and WKY-TRF groups from 8 to 24 weeks of age ($P > 0.05$), as shown in Fig 1. In addition, the SBP, DBP, and MBP of rats in the SHR-TRF group decreased by 25.740 ± 12.590 mmHg, 13.190 ± 11.380 mmHg, and 17.370 ± 10.830 mmHg, respectively, at 24 weeks of age compared with those at 8 weeks (Table 2, Fig 2).

## 3.2 Comparison of gross cardiac appearance, Body Weight (BW), HW, and HW/BW levels between groups

Gross cardiac appearance BW, HW, and HW/BW were compared to assess heart size and weight. Heart size was larger in both SHR-NON-FASTING and SHR-TRF groups compared to the WKY-NON-FASTING and WKY-TRF groups. However, the heart size in the SHR-TRF group was smaller than in the SHR-NON-FASTING group, yet it was still significantly larger than that in the WKY-NON-FASTING and WKY-TRF groups. No significant difference in heart size was observed between the WKY-NON-FASTING and WKY-TRF groups. BW was significantly lower in the SHR-NON-FASTING group compared to both the WKY-NON-FASTING and WKY-TRF groups ($P < 0.05$). BW was also significantly lower in the SHR-TRF group compared to both the SHR-NON-FASTING ($P < 0.001$) and WKY-NON-FASTING groups ($P < 0.05$), but there was no significant difference in BW between the SHR-TRF and WKY-TRF groups ($P > 0.05$). Moreover, BW was significantly lower in the WKY-TRF group compared to the WKY-NON-FASTING group ($P < 0.001$).

HW was significantly higher in the SHR-NON-FASTING group compared to both the WKY-NON-FASTING and WKY-TRF groups ($P < 0.001$). In contrast, HW was significantly lower in the SHR-TRF group compared to the SHR-NON-FASTING group ($P < 0.01$); however, still significantly higher than in both WKY-NON-FASTING ($P < 0.01$) and WKY-TRF groups ($P < 0.001$). There was no statistical significance in the HW between the WKY-NON-FASTING and WKY-TRF groups ($P > 0.05$).

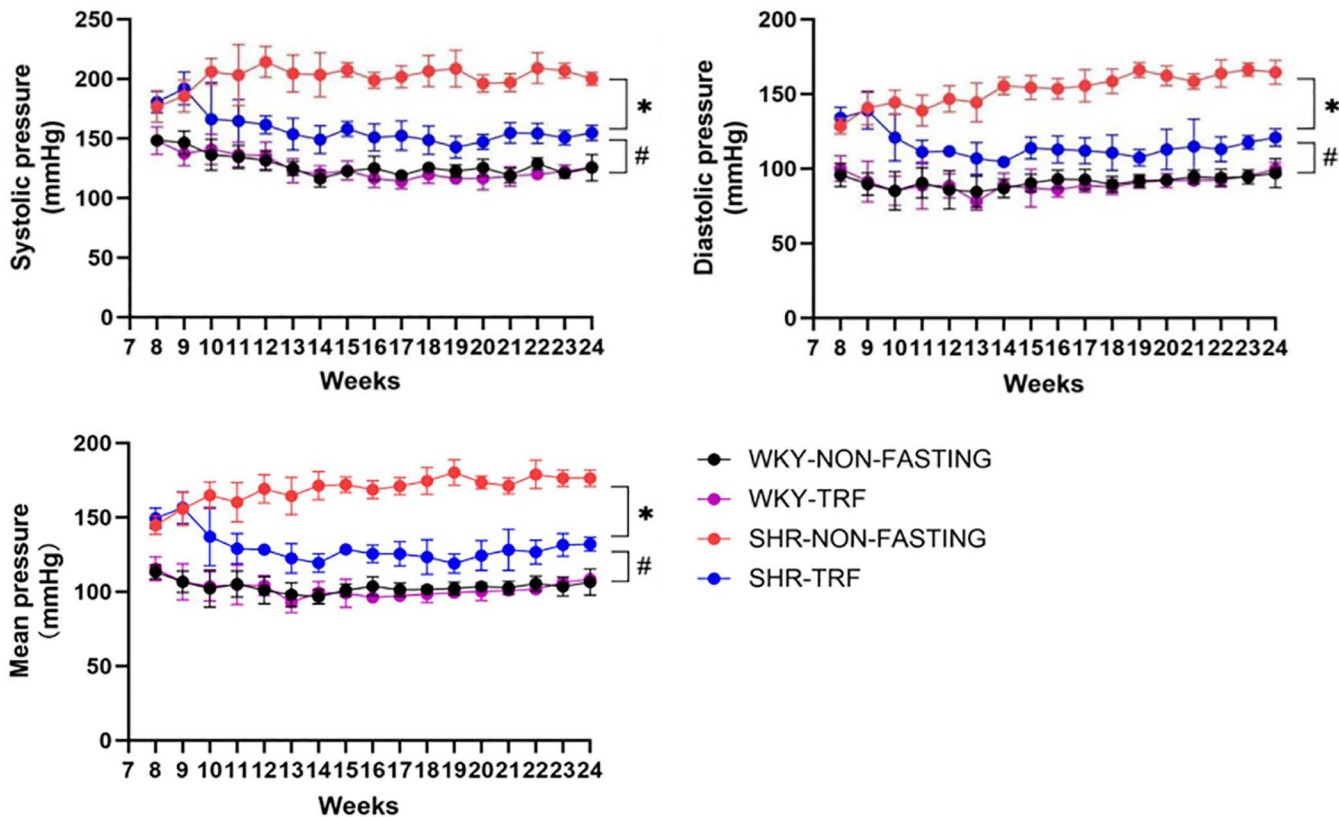

**Fig 1. Continuous BP Monitoring for 16weeks in 4 Groups.** *: From 10 week to 24 week for each node, SHR-TRF group compared with SHR-NON-FASTING group, P < 0.05; #: From 10 week to 24week for each node, SHR-TRF group compared with WKY-NON-FASTING and WKY-TRF groups, P < 0.05.

HW/BW levels were significantly higher in the SHR-NON-FASTING group compared to both the WKY-NON-FASTING ($P < 0.001$) and WKY-TRF groups ($P < 0.05$). HW/BW levels were significantly higher in the SHR-TRF group compared to both the WKY-NON-FASTING ($P < 0.001$) and WKY-TRF groups ($P < 0.01$). No statistical differences in HW/BW levels were observed between the WKY-NON-FASTING and WKY-TRF groups as well as between the SHR-NON-FASTING and SHR-TRF groups ($P > 0.05$)(Table 3. and Fig 3).

### 3.3 Comparison of HE staining between groups

In the WKY-NON-FASTING and WKY-TRF groups, cardiomyocytes were regularly arranged, with normal cell size, and no obvious pathological changes were observed. The SHR-NON-FASTING group showed a disorderly arrangement of cardiomyocytes, with a markedly larger cross-sectional area, and remarkable pathological changes were observed. In the SHR-TRF group, the arrangement of cardiomyocytes remained regular, the cross-sectional

**Table 2. Comparison of BP levels between 8 and 24 weeks of age in the SHR-TRF group.**

| BP | 8 weeks of age($n = 6$) | 24 weeks of age($n = 6$) | $t$ | $P$ |
|---|---|---|---|---|
| SBP(mmHg) | 180.400 ± 9.010 | 154.70 ± 6.371 | 5.006 | 0.004 |
| DBP(mmHg) | 134.000 ± 7.123 | 120.900 ± 5.829 | 2.837 | 0.036 |
| MBP(mmHg) | 149.500 ± 6.869 | 132.100 ± 4.767 | 3.927 | 0.011 |

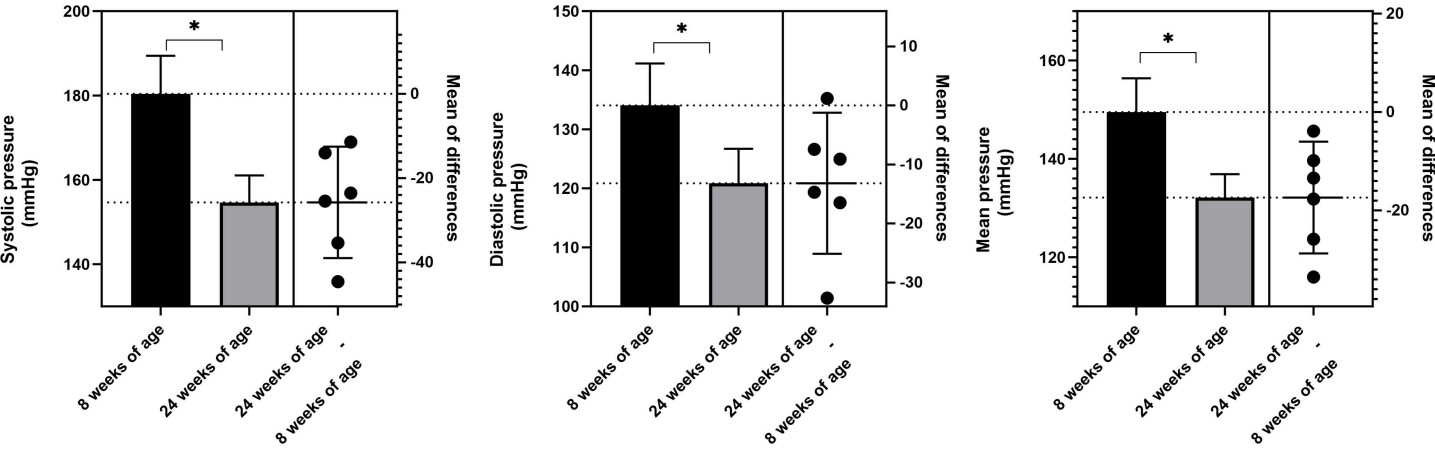

**Fig 2. Comparison of BP levels between 8 and 24 weeks of age in the SHR-TRF group \*: P < 0.05.**

**Table 3. Comparison of BW, HW, HW/BW levels between groups.**

| Groups | BW (g) | HW (g) | HW/BW |
|---|---|---|---|
| WKY-NON-FASTING($n=6$) | 383.167 ± 22.569 | 1.083 ± 0.066 | 0.283 ± 0.024 |
| WKY-TRF($n=6$) | 292.500 ± 26.403\*\*\* | 1.027 ± 0.127 | 0.355 ± 0.059 |
| SHR-NON-FASTING($n=6$) | 333.333 ± 25.936 \*\&\ | 1.468 ± 0.116 \*\*\*\*&&& | 0.444 ± 0.639\*\*\*& |
| SHR-TRF($n=6$) | 274.833 ± 44.364 \*\*\*% | 1.274 ± 0.111 \*\*\*\*&&% | 0.477 ± 0.103\*\*\*\*&& |

\*Compared with WKY-NON-FASTING group, $P<0.05$; \*\*: Compared with WKY-NON-FASTING group, $P<0.01$;

\*\*\*Compared with WKY-NON-FASTING Group, $P<0.001$; &: Compared with WKY-TRF group, $P<0.05$;

&&Compared with WKY-TRF group, $P<0.001$; &&&: Compared with WKY-TRF group, $P<0.001$;

%Compared with SHR-NON-FASTING group, $P<0.01$.

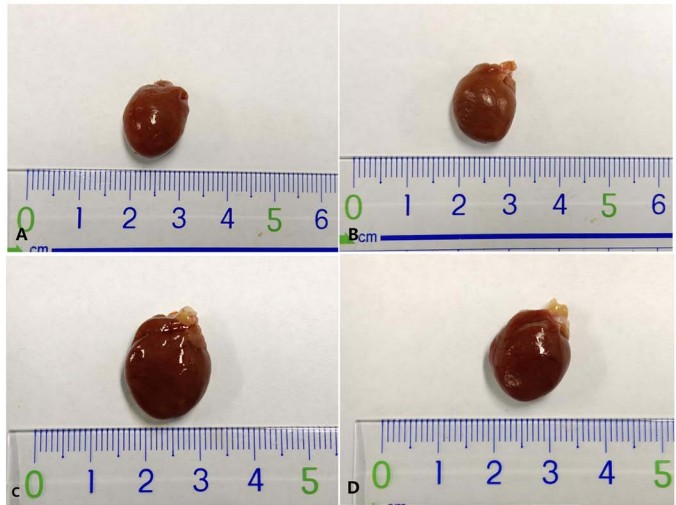

**Fig 3. Gross cardiac appearance in 4 groups.** (A) WKY-NON-FASTING Group; (B) WKY-TRF Group; (C) SHR-NON-FASTING Group; (D) SHR-TRF Group.

area of cells increased, and the pathological changes were less severe than those in the SHR-NON-FASTING group (Fig 4).

### 3.4 Comparison of ANP and BNP mRNA expression levels in LV tissue between groups

ANP and BNP mRNA expression levels were significantly higher in the SHR-NON-FASTING group compared to both the WKY-NON-FASTING and the WKY-TRF groups ($P < 0.001$). ANP and BNP mRNA expression levels were significantly lower in the SHR-TRF group compared to the SHR-NON-FASTING group but significantly higher in both the WKY-NON-FASTING and WKY-TRF groups ($P < 0.001$). BNP mRNA expression levels were significantly higher in the WKY-TRF group compared to the WKY-NON-FASTING group ($P < 0.05$). There was no statistical significance in the ANP expression levels between the WKY-NON-FASTING and WKY-TRF groups ($P > 0.05$) (Table 4, Fig 5).

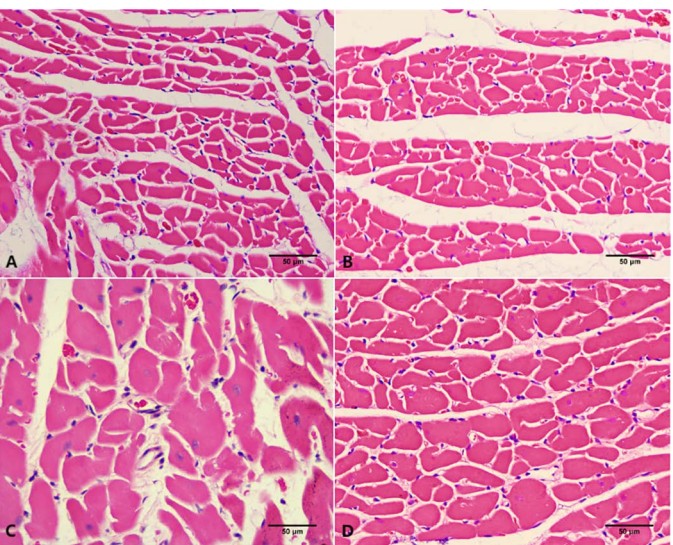

**Fig 4. HE staining of LV tissues in 4 groups.** (A) WKY-NON-FASTING Group; (B) WKY-TRF Group; (C) SHR-NON-FASTING Group; (D) SHR-TRF Group.

**Table 4. Comparison of ANP and BNP mRNA expression level in LV tissue between groups.**

| Groups | ANP mRNA expression level | BNP mRNA expression level |
|---|---|---|
| WKY-NON-FASTING(n=6) | $0.970 \pm 0.055$ | $1.041 \pm 0.085$ |
| WKY-TRF(n=6) | $0.918 \pm 0.055$ | $0.928 \pm 0.057$ * |
| SHR-NON-FASTING(n=6) | $3.008 \pm 0.100$ **& | $2.327 \pm 0.121$ **& |
| SHR-TRF(n=6) | $2.017 \pm 0.077$ **&% | $2.090 \pm 0.050$ **&% |

*: Compared with WKY-NON-FASTING group. $P < 0.05$; **: Compared with WKY-NON-FASTING Group, $P < 0.001$;

&: Compared with WKY-TRF group, $P < 0.05$; %:Compared with SHR-NON-FASTING group, $P < 0.001$.

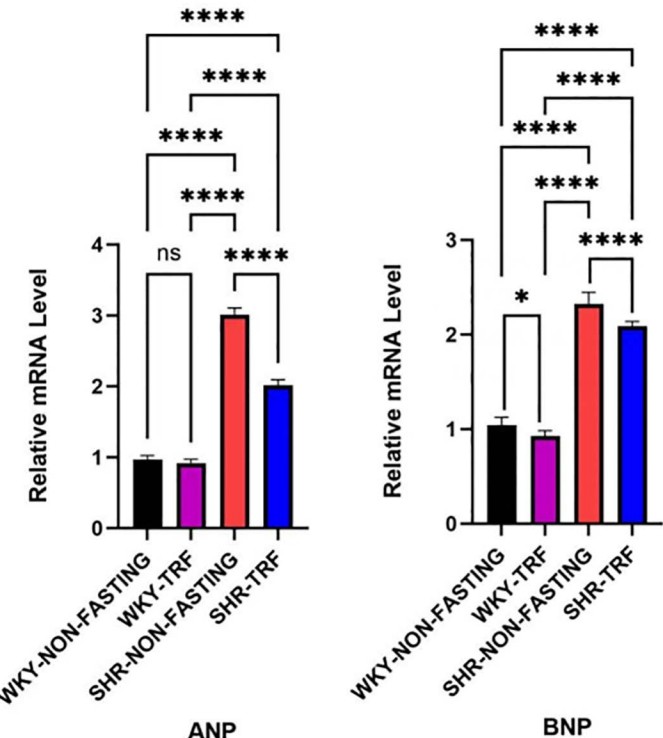

**Fig 5. Comparison of ANP and BNP mRNA expression level in LV tissue between groups.** (****: $P < 0.001$; *: $P < 0.05$; ns: No statistical difference, $P > 0.05$).

## 3.5 Comparison of ACE, Ang-II, and AT1 expression levels in serum and LV tissues between groups

The expression levels of ACE, Ang-II, and AT1 in the serum and LV tissues were significantly higher in the SHR-NON-FASTING group compared to both the WKY-NON-FASTING and the WKY-TRF groups ($P < 0.001$). These expression levels were significantly lower in the SHR-TRF group compared to that in the SHR-NON-FASTING group, although significantly higher than in both the WKY-NON-FASTING and WKY-TRF group ($P < 0.001$). A similar trend was observed in the serum-ACE, LV tissue-ACE, as well as serum-AT1 and LV tissue-AT1 expression levels, which were significantly higher in the WKY-TRF group compared to the

**Table 5. Comparison of ACE, Ang-II and AT-1 expression levels in serum between groups.**

| Groups | serum-ACE | serum-Ang-II | serum-AT1 |
|---|---|---|---|
| WKY-NON-FASTING($n = 6$) | $2.092 \pm 0.228$ | $152.270 \pm 23.049$ | $476.537 \pm 64.035$ |
| WKY-TRF($n = 6$) | $1.789 \pm 0.282$ * | $138.897 \pm 21.331$ | $425.796 \pm 59.655$ * |
| SHR-NON-FASTING($n = 6$) | $5.528 \pm 0.450$ **& | $314.214 \pm 33.149$ ** & | $769.778 \pm 79.976$ **& |
| SHR-TRF($n = 6$) | $4.005 \pm 0.393$ **&% | $277.825 \pm 21.929$ **&% | $680.426 \pm 89.806$ **&% |
| $F$ | 451.978 | 218.700 | 87.144 |
| $P$ | <0.001 | <0.001 | <0.001 |

*: Compared with WKY-NON-FASTING group, $P < 0.05$; **: Compared with WKY-NON-FASTING Group, $P < 0.001$;

&: Compared with WKY-TRF group, $P < 0.05$; %: Compared with SHR-NON-FASTING group, $P < 0.001$;

WKY-NON-FASTING group ($P < 0.05$). No significant difference was observed in serum-Ang-II and LV tissue-Ang-II expression levels between the WKY-NON-FASTING and WKY-TRF groups ($P > 0.05$) (Table 5, Table 6, Fig 6).

## 4 Discussion

This study found that BP, including SBP, DBP, and MBP, significantly decreased in SHR after 2 weeks of TRF intervention. In contrast, BP levels of SHR that did not receive TRF intervention continued to rise. Therefore, the antihypertensive effect of TRF was evident after 2 weeks. At this time, BP levels in SHR receiving TRF intervention were lower than those that did not receive TRF intervention; however, BP remained significantly higher than normal. This indicates that while TRF effectively lowers BP, its effect is limited, and this difference remained evident throughout the entire TRF intervention period. This result differs from SHI's study [22], which showed that ADF intervention reduced BP to normal levels in spontaneously hypertensive stroke-prone (SHR-SP) rats. This discrepancy may be due to the markedly shorter fasting duration in TRF compared to ADF, suggesting that different IF modalities may vary in their BP-lowering effects. Additionally, SHI's study initiated ADF intervention at 5 weeks of age without measuring BP levels before the intervention, and there was no significant decrease in BP levels in SHR-SP receiving ADF intervention [22]. Combined with the previously reported physiological characteristics of SHR [23], we, therefore, hypothesize that the earlier ADF intervention in SHI's study may have occurred before the development of hypertension, whereas in our study, the intervention was initiated after the development of hypertension. This may be the second reason for the inconsistency, as early intervention is generally associated with better outcomes in various diseases [24–26].

As a dietary intervention, TRF could be considered for long-term application in patients' daily lives to provide additional benefits. However, previous animal studies on IF and BP levels typically involved intervention periods of only a few weeks [9,16,22,27], leaving the long-term effects of IF on BP unclear. Our study addressed this gap, revealing that over time, from 3 weeks of TRF intervention to 16 weeks when the intervention ended, the BP levels of SHR receiving TRF remained lower than those of SHR without the intervention, although still significantly higher than normal levels. This finding suggested that the BP-lowering effect of TRF neither intensified nor diminished with prolonged intervention. Furthermore, TRF had no significant effect on normal BP levels, consistent with SHI's findings [22]. Due to the presence of spontaneous hypertension, SHR exhibited differences in physiological and metabolic mechanisms compared to WKY rats with normal BP levels. As a result, TRF intervention appeared to reduce BP levels by improving these physiological abnormalities. However, in WKY rats

**Table 6. Comparison of ACE, Ang-II and AT-1 expression levels in LV tissue between groups.**

| Groups | LV tissue-ACE | LV tissue-Ang-II | LV tissue-AT1 |
|---|---|---|---|
| WKY-NON-FASTING($n = 6$) | $3.698 \pm 0.369$ | $183.833 \pm 24.603$ | $388.815 \pm 42.794$ |
| WKY-TRF($n = 6$) | $3.335 \pm 0.324$ * | $168.648 \pm 25.635$ | $349.926 \pm 24.603$ * |
| SHR-NON-FASTING($n = 6$) | $7.530 \pm 0.659$ ***& | $379.944 \pm 28.654$ ***& | $708.907 \pm 59.909$ ***& |
| SHR-TRF($n = 6$) | $6.538 \pm 0.367$ ***&% | $331.426 \pm 45.022$ ***&% | $646.870 \pm 76.402$ ***&% |
| F | 382.802 | 195.554 | 197.991 |
| P | <0.001 | <0.001 | <0.001 |

*Compared with WKY-NON-FASTING group, $P < 0.05$; **: Compared with WKY-NON-FASTING Group, $P < 0.001$;

&Compared with WKY-TRF group, $P < 0.05$; %: Compared with SHR-NON-FASTING group, $P < 0.001$;

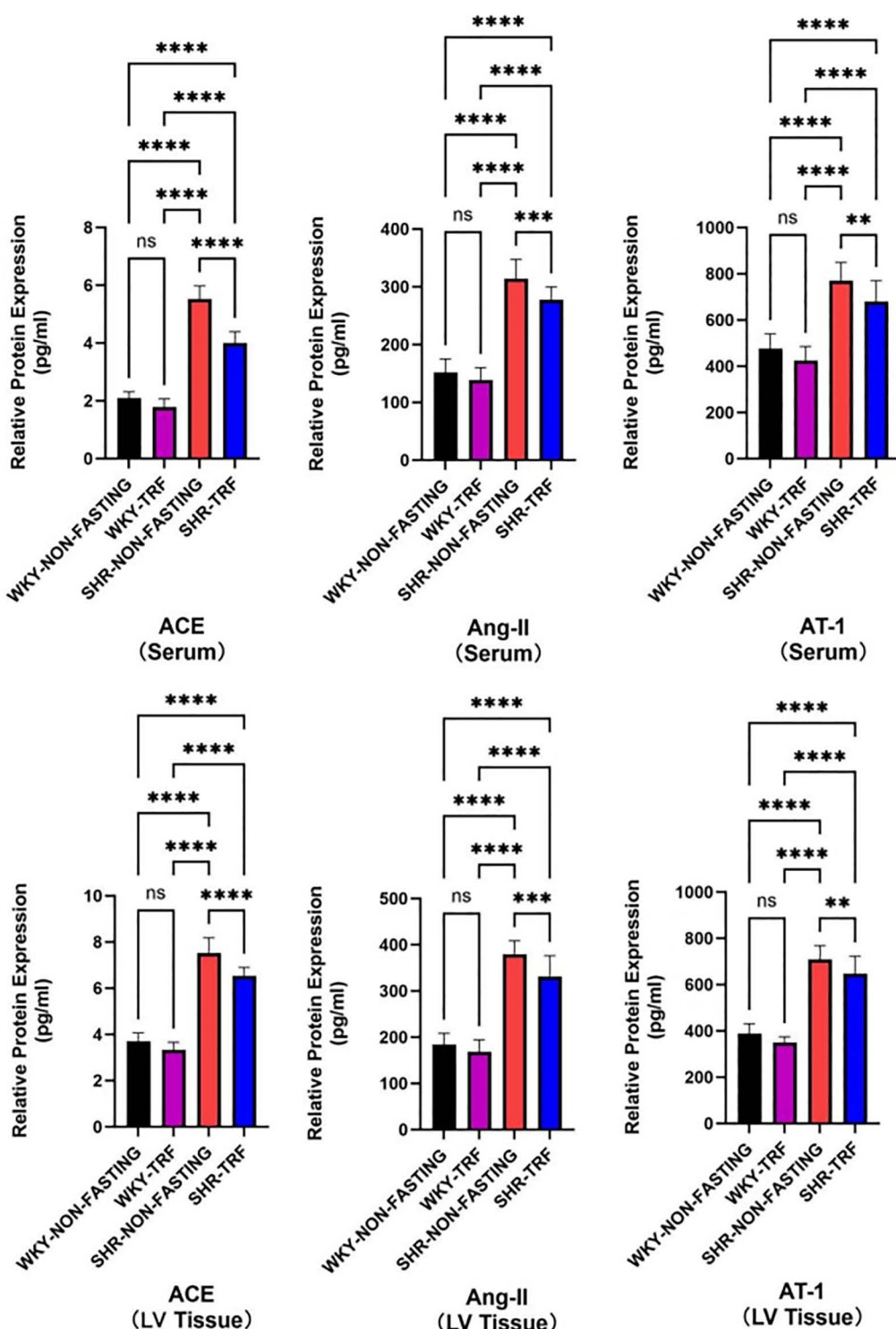

**Fig 6. Comparison of ACE, AngII and AT1 mRNA expression levels in serum and LV tissues between groups.** (****: $P < 0.001$; **: $P < 0.05$; ns: No statistical difference, $P > 0.05$).

with normal BP levels, the effect of TRF intervention was limited, suggesting minimal impact on normal BP levels. Hypertensive individuals appeared to be more sensitive to the regulatory effects of TRF, whereas the physiological mechanisms in normotensive individuals did not require further intervention.

The control of BW plays a critical role in hypertension treatment [28,29]. Previous studies [11,30–32] have demonstrated that IF can effectively reduce BW in various animal models. Our study corroborates these findings by showing that TRF effectively reduced BW in both SHR and WKY rats. Owing to hypertension-induced hypermetabolism, SHR initially had lower BW compared to WKY rats, and TRF further reduced BW in SHR, potentially limiting its applicability in patients with already low BW. At the end of the 16-week TRF intervention, BW levels of SHR and WKY rats receiving TRF were at the same level, indicating a steady reduction in BW rather than a transitional manner. These results are consistent with the findings of SHI et al. [22] on the effects of ADF on BW in SHR-SP, suggesting similar impacts of ADF and TRF on BW reduction. Moreover, TRF decreased HW in SHR but did not significantly affect the HW/BW ratio, possibly due to a greater reduction in BW than in HW.

A previous study [9] showed that ADF improves the cardiac structure of mice with metabolic cardiomyopathy induced by a high-fat or high-sugar diet. Our findings similarly indicate that TRF reduces overall heart and cardiomyocyte size in SHR, potentially improving hypertension-induced structural heart damage. Given the close relationship between cardiac structure and function, ANP and BNP, which are peptide hormones secreted by the atria and ventricles, are important modulators of the cardiovascular system, accurately reflecting cardiac function, and play important roles in the regulation of BP and fluid balance [33,34]. We demonstrated that TRF reduced the mRNA expression levels of ANP and BNP in LV tissues of SHR but did not reduce them to normal levels, indicating potential improvement in hypertension-induced impairment of cardiac function. TRF also reduced BNP levels in LV tissues of WKY rats, even in the absence of any disease, suggesting potential cardio-protective effects against age-related cardiac decline [35,36].

The activity of RAS is closely related to hypertension and hypertension-induced cardiac remodeling [37,38]. Camelo's study [6] demonstrated that ADF for 4 weeks lowered BP levels by inhibiting local RAS activity in the heart of mice with metabolic heart disease. In contrast, our study used the SHR model, a widely recognized model for essential hypertension, and administered a longer intervention of TRF for 16 weeks. We demonstrated that TRF reduces ACE, Ang-II, and AT1 expression levels in LV tissues and serum. This reduction suggests that TRF inhibits circulating and local RAS, providing a dual mechanism for lowering BP and improving cardiac structure and function. Extending the duration of TRF to 16 weeks demonstrates its sustained and long-term efficacy in BP reduction and cardiac protection. Moreover, the shorter fasting period required for TRF compared to ADF makes it a more feasible and clinically translatable dietary intervention. The relationship between RAS activity and ANP and BNP levels supports their synergistic effects in volume load reduction, vasodilation, and cardio-protection [39–41]. Additionally, we found that TRF inhibited RAS activity in the serum and LV tissue of WKY rats, which may be one of the reasons for the reported cardioprotective effects. However, TRF had no significant effect on BP levels in WKY, likely due to the influence of multiple neurohumoral factors in addition to RAS regulation [42,43]. Even if RAS is inhibited, BP may still be in the normal range, suggesting that TRF has a role in BP homeostasis maintenance. These findings underscore the potential of TRF as a practical and durable therapeutic strategy for managing hypertension and improving cardiovascular outcomes.

As far as we know, our study is the first to investigate TRF' s effects on RAS activity, as well as cardiac structure and function simultaneously. Despite limitations such as the lack

of combined drug interventions or RAS inhibitors and a relatively small sample size, our findings provide significant theoretical support for future research on TRF, hypertension, and hypertension-induced cardiac remodeling. Importantly, our results emphasize the potential of TRF as a therapeutic approach, highlighting its role in refining strategies for managing cardiac remodeling. These insights offer new perspectives and practical guidance for clinical applications in the treatment and management of hypertension. Building on our findings, future research should prioritize expanding the study larger animal cohorts and diverse hypertension models to validate the reproducibility and robustness of these results across varied experimental conditions. Furthermore, exploring the integration of TRF with RAS inhibitors or other antihypertensive agents in preclinical models may reveal potential synergistic effects, laying the groundwork for combination strategies in hypertension management.

## Supporting information

**S1 Appendix. The raw data of all data presented.** The raw data includes BP monitoring data, heart size and weight measurements, ELISA data, and PCR data.
(XLSX).

## Author contributions

**Conceptualization:** Xin Yi, Jie Yan, Ummi Nadira Daut.

**Data curation:** Xin Yi, Weijing Sun.

**Formal analysis:** Xin Yi, Ummi Nadira Daut.

**Funding acquisition:** Xin Yi.

**Investigation:** Xin Yi.

**Methodology:** Xin Yi, Razif Abas, Canzhang Liu, Ummi Nadira Daut.

**Project administration:** Xin Yi, Ummi Nadira Daut.

**Resources:** Xin Yi.

**Software:** Xin Yi, Chongshuang Yang, Teng Gao.

**Supervision:** Xin Yi, Razif Abas, Ummi Nadira Daut.

**Validation:** Xin Yi, Razif Abas, Ummi Nadira Daut.

**Visualization:** Xin Yi, Ummi Nadira Daut.

**Writing – original draft:** Xin Yi, Raja Abdul Wafy Raja Muhammad Rooshdi.

**Writing – review & editing:** Xin Yi, Razif Abas, Ummi Nadira Daut.

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
