## [Decision Letter · Decision Letter 0]

19 Dec 2024

PONE-D-24-49902Time-Restricted Feeding Reduced Blood Pressure and Improved Cardiac Structure and Function by Regulating Both Circulating and Local Renin-Angiotensin Systems in Spontaneously Hypertensive Rat ModelPLOS ONE

Dear Dr. YI,

Thank you for submitting your manuscript to PLOS ONE. After careful consideration, we feel that it has merit but does not fully meet PLOS ONE’s publication criteria as it currently stands. Therefore, we invite you to submit a revised version of the manuscript that addresses the points raised during the review process.

We look forward to receiving your revised manuscript.

Kind regards,

Sheryar Afzal, Ph.D

Academic Editor

PLOS ONE

Journal Requirements:

2. To comply with PLOS ONE submissions requirements, in your Methods section, please provide additional information regarding the experiments involving animals and ensure you have included details on methods of sacrifice.

3. We note that your Data Availability Statement is currently as follows: “All relevant data are within the manuscript and in Supporting Information files.”

Additional Editor Comments:

ONE-PONE-D-24-49902

Time-Restricted Feeding Reduced Blood Pressure and Improved Cardiac Structure and Function by Regulating both Circulating and Local Renin-Angiotensin Systems in Spontaneously Hypertensive Rat Model

PLOS ONE

Dear Authors,

Thank you for submitting your manuscript to PLOS ONE. After careful consideration, we feel that it has merit but does not fully meet journals publication criteria as it currently stands. Therefore, I recommend addressing all the important points raised by the reviewers in their report and addresses them point by point in the revised version of the manuscript for better understanding of the readers,

Kindly follow the proper guidelines for resubmitting the revised version along rebuttal letter for complete understanding of the reviewers during 2nd round of review process, before we proceed further in the publication process.

We look forward to receiving your revised manuscript.

Kind regards and best wishes

Sheryar Afzal, Ph.D.

Academic Editor

PLOS ONE

Reviewers' comments:

Reviewer's Responses to Questions

**Comments to the Author**

1. Is the manuscript technically sound, and do the data support the conclusions?

Reviewer #1: Yes

Reviewer #2: Yes

2. Has the statistical analysis been performed appropriately and rigorously? 

Reviewer #1: Yes

Reviewer #2: Yes

3. Have the authors made all data underlying the findings in their manuscript fully available?

Reviewer #1: Yes

Reviewer #2: Yes

4. Is the manuscript presented in an intelligible fashion and written in standard English?

Reviewer #1: Yes

Reviewer #2: Yes

5. Review Comments to the Author

Reviewer #1: In this paper Xin and co-authors determined to investigate whether time-restricted feeding (TRF) can reduce blood pressure (BP) and improve cardiac structure and function in spontaneously hypertensive rats (SHRs) by regulating the renin-angiotensin system (RAS). The outcomes of this study could be beneficial in terms of clinical practice and novel therapeutic strategies concerning hypertension.

1. Input some statistic figure for the hypertension as global health issue

2. IF stands for what?, abbreviation should be put at the first time when it is used.

3. In text, it mentioned, “RAS is a central component of the pathological process of hypertension and hypertension-induced cardiac remodeling and is a key therapeutic target”… Include some pathological evidence on RAS that related to hypertension after line 50.

For the laboratory animals, authors should present the animal sampling, or sample size calculation, inclusion and exclusion criteria, and or be included as supplementary files.

1. Please add the citation to support the sentences at line 214-216 regarding the SHI’S study of ADF intervention.

2. Please add future perspective of the study.

References are not properly written in the bibliography section as per journals criteria and style.

Reviewer #2: The study entitled "Time-Restricted Feeding Reduced Blood Pressure and Improved Cardiac Structure

and Function by Regulating Both Circulating and Local Renin-Angiotensin Systems in Spontaneously Hypertensive Rat Model" is interesting and well-designed. The authors tried to investigate the time-restricted feeding (TRF) effect on blood pressure reduction and associated improvement in cardiac anatomy and physiology in SHRs via RAAS regulation.

The study is interesting; however, the following points must be cleared for a better understanding of the study data and before proceeding for publication purposes.

1. the current statistical global update for hypertension and its associated comorbidities

2. All abbreviations must be defined first before being used in the text. Numerous terms are not explained properly while using abbreviations.

3. The mechanism of RAAS’s pathological association with hypertension and cardiac remodeling needs further explanation with the help of recent literature for a better understanding of the hypothesis and objectives of the study.

4. Did the authors observe any inclusion and exclusion criteria for the selection of animals? SHRs belong to the genetic model of hypertension and are age-dependent, competing with the criteria for hypertension.

5. If yes, please also provide complete details for animal grouping animals, acclimatization period, and environment

6. To improve the discussion further, compare the data findings with the recent literature on parallel lines to support the conclusions and study hypothesis.

7. A few studies are not properly cited in the discussion part, which creates ambiguity while understanding the study findings.

8. An example is "in lines 221 to 223, there is no supporting evidence from previous work on, if any, for the limited efficiency of TRF in prolonged intervention.

9. In conclusion, elaborate clearly that the study's outcome will assist in highlighting and identifying the therapeutic strategies for cardiac remodelling.

10. Some references are not properly cited. The journal editorial office must check the style of the references before final acceptance of the manuscript.

6. PLOS authors have the option to publish the peer review history of their article (what does this mean?). If published, this will include your full peer review and any attached files.

Reviewer #1: **Yes: **Dr. Wu Yuan Seng

Reviewer #2: **Yes: **ALI ATTIQ

---

## [Author Response · Author response to Decision Letter 0]

18 Jan 2025

Please see the 'Response to Reviewers' document

---

## [Decision Letter · Decision Letter 1]

2 Mar 2025

Time-Restricted Feeding Reduced Blood Pressure and Improved Cardiac Structure and Function by Regulating Both Circulating and Local Renin-Angiotensin Systems in Spontaneously Hypertensive Rat Model

PONE-D-24-49902R1

Dear Dr. Ummi Nadira Daut

We’re pleased to inform you that your manuscript has been judged scientifically suitable for publication and will be formally accepted for publication once it meets all outstanding technical requirements.

Within one week, you’ll receive an e-mail detailing the required amendments. When these have been addressed, you’ll receive a formal acceptance letter, and your manuscript will be scheduled for publication.

An invoice will be generated when your article is formally accepted. Please note that if your institution has a publishing partnership with PLOS and your article meets the relevant criteria, all or part of your publication costs will be covered. Please make sure your user information is up-to-date by logging into Editorial Manager at Editorial Manager® and clicking the ‘Update My Information' link at the top of the page. If you have any questions relating to publication charges, please contact our Author Billing department directly at authorbilling@plos.org.

Kind regards,

Sheryar Afzal, Ph.D.

Academic Editor

PLOS ONE

Additional Editor Comments (optional):

Reviewers' comments:

Reviewer's Responses to Questions

**Comments to the Author**

1. If the authors have adequately addressed your comments raised in a previous round of review and you feel that this manuscript is now acceptable for publication, you may indicate that here to bypass the “Comments to the Author” section, enter your conflict of interest statement in the “Confidential to Editor” section, and submit your "Accept" recommendation.

Reviewer #2: All comments have been addressed

2. Is the manuscript technically sound, and do the data support the conclusions?

Reviewer #2: Yes

3. Has the statistical analysis been performed appropriately and rigorously? 

Reviewer #2: Yes

4. Have the authors made all data underlying the findings in their manuscript fully available?

Reviewer #2: Yes

5. Is the manuscript presented in an intelligible fashion and written in standard English?

Reviewer #2: Yes

6. Review Comments to the Author

Reviewer #2: (No Response)

7. PLOS authors have the option to publish the peer review history of their article (what does this mean?). If published, this will include your full peer review and any attached files.

Reviewer #2: **Yes: **ALI ATTIQ

---

## [Editor Report · Acceptance letter]

PONE-D-24-49902R1

PLOS ONE

Dear Dr. YI,

I'm pleased to inform you that your manuscript has been deemed suitable for publication in PLOS ONE. Congratulations! Your manuscript is now being handed over to our production team.

Kind regards,

on behalf of

Dr. Sheryar Afzal

Academic Editor

PLOS ONE